# Diabetes Capabilities for the Healthcare Workforce Identified via a 3-Staged Modified Delphi Technique

**DOI:** 10.3390/ijerph19021012

**Published:** 2022-01-17

**Authors:** Giuliana Murfet, Joan Ostaszkiewicz, Bodil Rasmussen

**Affiliations:** 1School of Nursing and Midwifery, Deakin University, 221 Burwood Highway, Burwood, VIC 3125, Australia; bodil.rasmussen@deakin.edu.au; 2Diabetes Centre, Tasmanian Health Service, Burnie, TAS 7250, Australia; 3School of Public Health, University of Technology Sydney, 15 Broadway, Ultimo, NSW 2007, Australia; 4National Aging Research Institute, Royal Melbourne Hospital, Parkville, VIC 3050, Australia; j.ostaszkiewicz@nari.edu.au; 5Centre for Quality and Patient Safety Research, Institute for Health Transformation, Deakin University, 1 Geringhap Street, Geelong, VIC 3220, Australia; 6Western Health Partnership, 176 Furlong Road, St Albans, VIC 3021, Australia; 7Department of Public Health, University of Copenhagen, Nørregade 10, DK-1017 Copenhagen, Denmark; 8Faculty of Health Sciences, University of Southern Denmark, Campusvej 55, DK-5230 Odense, Denmark

**Keywords:** diabetes, healthcare workforce, *Delphi* technique, capabilities, workforce capacity

## Abstract

Consumers access health professionals with varying levels of diabetes-specific knowledge and training, often resulting in conflicting advice. Conflicting health messages lead to consumer disengagement. The study aimed to identify capabilities required by health professionals to deliver diabetes education and care to develop a national consensus capability-based framework to guide their training. A 3-staged modified *Delphi* technique was used to gain agreement from a purposefully recruited panel of Australian diabetes experts from various disciplines and work settings. The Delphi technique consisted of (Stage I) a semi-structured consultation group and pre-*Delphi* pilot, (Stage II) a 2-phased online *Delphi* survey, and (Stage III) a semi-structured focus group and appraisal by health professional regulatory and training organisations. Descriptive statistics and central tendency measures calculated determined quantitative data characteristics and consensus. Content analysis using emergent coding was used for qualitative content. Eighty-four diabetes experts were recruited from nursing and midwifery (*n* = 60 [71%]), allied health (*n* = 17 [20%]), and pharmacy (*n* = 7 [9%]) disciplines. Participant responses identified 7 health professional practice levels requiring differences in diabetes training, 9 capability areas to support care, and 2 to 16 statements attained consensus for each capability—259 in total. Additionally, workforce solutions were identified to expand capacity for diabetes care. The rigorous consultation process led to the design and validation of a *Capability Framework for Diabetes Care* that addresses workforce enablers identified by the *Australian National Diabetes Strategy*. It recognises diversity, creating shared understandings of diabetes across health professional disciplines. The findings will inform diabetes policy, practice, education, and research.

## 1. Introduction

Over a quarter of Australians accessing a healthcare service on any given day have diabetes [1,2]. These consumers receive education and care from health professionals with varying levels of diabetes experience in all areas of Australia. However, health professionals’ approach to delivering information and the information provided can engage or disengage the person accessing services [3,4]. When health messaging is inconsistent, people living with diabetes tend to disengage from essential health services [3,4]. Reduced access to care by diabetes-competent health professionals increases the risk of preventable costly diabetes-related complications [5,6] that make people living with diabetes retire early, fall into income poverty, and omit care because of the cost, which increases healthcare needs [7].

By 2030, diabetes will be the leading cause of health condition burden in Australia [5], when over half of the primarily tertiary-based Credentialled Diabetes Educator^TM^ (CDE) workforce reaches retirement age [8]. In Australia, CDEs are health professionals who have met the requirements for the title by completing a postgraduate degree in diabetes, 1000 initial hours of diabetes practice, and a 6-months mentorship and have demonstrated ongoing annual participation in professional development within the specialty of diabetes education [9]. A mature and retiring diabetes-trained workforce will reduce the Australian health system’s capacity to deliver diabetes care and prevention over the next decade. Hence, to reduce the human and financial burden to people living with diabetes, their families, and the healthcare system, there is a need to ensure that all health workforce members have appropriate knowledge, skills, and attributes to care for them.

The Australian Commonwealth Government and States continue developing policies to fund exponential growth in diabetes-related healthcare needs [10,11,12]. The *Australian National Diabetes Strategy 2021–2030* identified the workforce as an enabler of access to equitable person-focused integrated quality care spanning the health continuum [13].

Internationally, it is recognised that this access will not be sustainable without increasing health professionals’ diabetes capabilities [14]. Australian policies to better support diabetes care include Medicare-incentivised models (publicly funded universal healthcare) for general medical practitioners to manage diabetes with generalists, primary healthcare nurses, and allied health professionals in a coordinated manner [10,11,12]. Driven by policy and employer expectations, primary health professionals must now undertake activities once performed by specialist CDEs [15,16]. However, primary healthcare professionals report being ill-prepared to deliver diabetes education without adequate diabetes capabilities internationally and in Australia [14,15,16,17,18].

Australia, like other countries, also has significant complexities and challenges with healthcare delivery, including multiculturalism, which requires that services and information to be tailored appropriately [19,20,21] and that inequities to healthcare in Indigenous people need to be addressed culturally [22,23,24]. Further, it is nuanced by geographical areas of significant remoteness [25], where rates of diabetes are high and the duration longer and healthcare access is limited [23,25,26].

Furthermore, the number and complexity of technologies and medicines used to manage diabetes have increased exponentially [27,28], which drive healthcare changes. Australia’s areas of remoteness, where extreme shortages of doctors and populations of extreme disadvantage exist [25,26,29], affect healthcare choices available to and made by consumers. For example, despite over a third of Australians with type 1 diabetes residing in areas of increasing remoteness [30], they account for less than 10% of the total insulin pump usage due to perceived lack of support [31,32]. The broader health professional workforce is not adequately prepared or resourced for these challenges [15,25].

Diabetes care is *everyone’s* business. An imperative exists to develop the future health workforce’s capacity to deliver quality primary and preventive diabetes care at a population level. Identifying ways to engage better and meet the diabetes training needs of health professionals working in all settings, particularly rural and very remote areas of Australia, is essential [13].

*Competency frameworks* are used in many countries to prepare health professionals; they are task-orientated tools that standardise technical skills in stable clinical situations [33]. However, existing competency frameworks designed to equip health professionals with the knowledge, skills, and attributes to provide diabetes care are inadequate [34]. They are unlikely to support the development of the workforce because no diabetes framework in Australia informs the *whole* workforce, nor are any capability-based. Finally, competency frameworks have little utility in an environment aimed at increasing scope of practice by creating a skilled, flexible, and innovative workforce [35], nor do they recognise the benefits of autonomous practices, which enable innovation [36].

*Capability-based learning* offers an alternative; while it encompasses competency, it extends beyond technical skills to emphasise the components of adaptability to change, lifelong learning, and self-efficacy [35,36,37]. To expand the workforce’s capacity to address future diabetes healthcare needs, understanding the diabetes capabilities required by health professionals is essential. Australia’s geographical nuances require a flexible, adaptable diabetes workforce with skills to practice at advanced and extended levels to provide appropriate safe diabetes care [25,26]. The current study aimed to develop a consensus capability framework to guide non-medical health professional training and development in delivering diabetes education and care across the Australian healthcare spectrum [34]. The research objectives were to:Identify and understand the different non-medical healthcare professional practice levels used to deliver diabetes care in Australia;Identify the capabilities required to deliver quality, safe diabetes education and care;Increase the health workforce’s capacity to manage diabetes.

## 2. Materials and Methods

### 2.1. Methodological Approach

A modified *Delphi* technique was used to conduct the study, administered in three stages (see Figure 1). The *Delphi* technique was modified by introducing a pre-*Delphi* consultation stage to understand workforce development issues better [38]. The *Delphi* technique offers the advantage of providing a systematic consensus-building methodology to identify healthcare capabilities or priorities [38,39,40]. It enables a group of ‘experts’ to explore complex issues and reach a consensus [40], which benefited this study as opinions about roles in diabetes are diverse [41]. *Delphi* features that ensured information was captured accurately from individual participants were anonymity, eliminating group pressure and the influence of dominant personalities, iterations reviews with controlled feedback, and quantified ranking of group responses [38,40].

### 2.2. Theoretical Framework

Several theories underpinned the study and the development of the capability framework structure. The framework’s focus was to increase workforce capacity in diabetes by guiding training for a flexible workforce. Theories were divided into three categories: clinical competence, expertise, and capability. Benner’s *stages of clinical competence*, Ericsson’s *theory of expertise*, and Sen’s *capability approach theory* were used as the theoretical framework to guide the research [42,43,44].

### 2.3. Research Characteristics and Reflexivity

As an instrument within the research process that can profoundly affect the research [45], G.M. describes her experiences that could have influenced the study in unknown ways. As a Nurse Practitioner and CDE with 30-years’ experience in clinical and management settings, G.M. has supported people who live with diabetes through assessment, diagnosis, prescribing, education, and research. G.M. has held leadership positions nationally as Board Director of the Australian Diabetes Educators Association (ADEA) that credentials diabetes educators and Diabetes Australia, a consumer advocacy organisation. G.M. approved the ADEA Indigenous Educational Pathways Project, among other projects. G.M. has represented diabetes and nursing nationally at Ministerial advisory groups and Government-funded Pharmaceutical Benefits Scheme reviews. The current study’s focus fitted with G.M’s keen interest for equitable access to high standards of safe quality care, regardless of the healthcare setting and consumer needs.

### 2.4. Setting

The setting was the Australian health system. The sampling population was nurses, midwives, pharmacists, and allied health professionals employed to deliver diabetes education and care with people living with diabetes in a clinical, management, or educational role. Additionally, academics in diabetes workforce training, research, and professional development were involved.

### 2.5. Sampling Strategy

A heterogeneous and representative sample was recruited by applying the proportions of health disciplines established through the ADEA 2017 member survey [8]. Seven health discipline organisations advertised the study openly via member communications to reduce bias, which included a Plain Language Statement.

The expert *Delphi* participants were recruited over 11 weeks by a combination of purposeful and snowballing sampling approaches. Data saturation was monitored to ensure an adequate sample size [46]. Recruitment stopped when participants’ feedback provided no new insights.

Benner’s theoretical stages of clinical competence, which has been validated in various adult learning settings with different disciplines [42], and Ericsson’s theory of expertise, which recognises the importance of routine *deliberate practice* [43], informed the inclusion criteria—to ensure sampling from expert diabetes health professionals and academics from various disciplines who deliver care, lectures or research in diabetes. Health professionals required more than five years post-registration experience and to be either:A CDE for five years or more;In a position whose focus of employment for the past five years or more was diabetes education and care or/and research.

### 2.6. Ethical Considerations

No participants’ names were recorded or was identifiable information shared to maintain confidentiality. A password-protected computer, survey platform, and email address were used to conduct the survey. Data exported to Excel was stored on Deakin University’s system, offering high-level security and anti-virus programs to maintain data integrity. Data will be destroyed after five years, following Deakin policy.

### 2.7. Data Collection Methods

Various data collection methods were used in the *Delphi* technique, which informed sequential steps. See Figure 2 for the chronology of the data collection methods.

#### 2.7.1. Stage I


*Pre-Delphi semi-structured consultation group*


The purpose of the pre-*Delphi* consultation group was to understand issues impacting diabetes workforce development. It was limited to 20 participants to collect quality dialogue and diverse opinions [47,48]. An advertisement including a *SurveyMonkey* link captured demographic data to ensure inclusion criteria and adequate breadth of disciplines were met.

An independent, experienced moderator led the audiotaped semi-structured discussion. The moderator followed a protocol, which included seven guiding questions regarding workforce issues, differences in health professional diabetes training and skills requirements, and risks, benefits, and barriers to non-medical prescribing. G.M. took contextual notes regarding the environment, participant engagement, and body language and summarised emerging issues from the conversation. Summaries were fed-back verbally at intervals during the discussion to ensure the interpretation was correct. Finally, the audiotape was transcribed verbatim.


*Pre-Delphi pilot*


The purpose of the pilot was to establish whether respondents interpreted the questions and instructions as intended, to test the features of Qualtrics’, the web-based platform, and to estimate the time taken to enhance the questionnaire’s face and content validity and trustworthiness. Seven purposefully sampled diverse diabetes experts participated, and Qualtrics was used to forward the questionnaire via email. Their feedback guided the amendments to features implemented to enable flexibility to support engagement. Next, Stage II commenced.

#### 2.7.2. Stage II

The purpose of Stage II was to address the study’s research questions and gain consensus on different framework aspects.

*Expert Advisory Group*: Stage II and III were guided by five independent researchers to support the validation and trustworthiness of results at data analysis points and ensure:No capability components were overlooked;Capabilities identified would allow workforce capacity growth;The volume of data collected during each method was managed;An accurate model describing practice levels was chosen.

The EAG members held expertise in qualitative and *Delphi* research, pregnancy, technology, emotional health, prescribing, and course development. They were all national leaders in diabetes education and care and research from nursing, dietetics, midwifery, and behavioural science (see Appendix A). The EAG provided objective evaluations through peer-debriefing and technical and explanatory guidance; consensus regarding analyses was based on the majority.

*Delphi survey Phase one:* The purpose of Phase one was to identify the health professional practice levels signifying a change in diabetes knowledge and skills required. An online questionnaire was emailed to participants, asking demographic characteristics and three focused questions (see Figure 2). Following analyses and EAG appraisal and consensus, a second-round questionnaire presented participants with four models describing different *diabetes practice levels* [34]. The four models’ focuses were: (i) nursing only, (ii) Government-registered health professionals only, (iii) multidisciplinary, and (iv) work-setting, i.e., tertiary, secondary, primary, and remote. Participants were invited to rank their preferred model from 1 (preferred) to 4 (least preferred).

*Delphi survey Phase two:* The purpose of Phase two was to identify the essential capabilities required for diabetes care and reach a consensus about the alignment of specific capability components for each diabetes practice level (see Figure 2) [34]. Following analyses and EAG peer review and consensus, participants were emailed a questionnaire with 2 to 16 capability components under each diabetes practice level. Following further analyses and EAG appraisal, the second-round questionnaire included a summary and the capability components that had not reached consensus. Participants were invited to re-rank the statements. Next, a draft *Capability Framework for Diabetes Care* (‘Capability Framework’) was prepared and confirmed by the EAG in preparation for Stage III.

#### 2.7.3. Stage III

*Focus group:* In Stage III, a focus group was convened to validate the draft ‘Capability Framework’ to ensure the information gathered was well-grounded and transferable to the real world. The group was smaller, as four to six participants maintained higher communication quality levels better for brainstorming [45]. The ‘Capability Framework’ was appraised for completeness and practicality and to identify limitations or gaps against the study’s aims [47]. Following minor amendments and EAG review, the draft framework was forwarded for stakeholder appraisal (see Figure 2).

*Stakeholder appraisal and feedback:* The final data collection activity was a stakeholder consultation to identify any issues impeding practice or safety and determine whether the ‘Capability Framework’ addressed the *Australian National Diabetes Strategy* 2021–2030 [13]. Sixteen Australian educational organisations were invited to appraise and provide feedback about the framework via a survey. The survey was open for 4 weeks on Qualtrics or hard copy. Questions focused on identifying whether the framework was easy to navigate and on clarifying whether any capability impeded practice or could impact safe practice. 

### 2.8. Data Analysis

Each data collection method informed the next; qualitative and quantitative data were collected.

#### 2.8.1. Qualitative Data Analysis

Content analyses were used to identify, analyse, categorise, and report patterns within qualitative data throughout the *Delphi* stages [49]. The study used an inductive approach and emergent coding to manage issues [50]. The following coding rules were applied before content analysis to maintain rigor, as coding is a reflective process [49]:The level of analysis used was word sense, phrase, or sentences of similar meanings;Flexibility in the coding process with no predefined number or list of concepts;Concepts would be coded for their content, not frequency;Concepts would be developed to align with study questions and aims as they emerged into categories.

G.M. established categories and sub-categories following the preliminary data examination after repeated listening to audio recordings and reading verbatim transcripts [50]. Word clouds identified the diabetes experts’ language used to inform titles and categories to reduce researcher impact. Repetition of similar words or meanings enabled data saturation to be achieved/identified. Key reoccurring sub-categories were appraised by T.D. and J.O. for accuracy. Then, G.M. reported on categories and patterns to the EAG, who cross-checked emerging categories for accuracy, relevancy, applicability, duplication, or need for further condensing. In Delphi survey Phase one, practice levels were defined, and a stage of *diabetes clinical competence* derived from participants words was allocated. Next, capabilities were given a definition, and their components were finalised, assessed against study aims, and critically reviewed until consensus was achieved.

#### 2.8.2. Quantitative Data Analyses

Descriptive statistics and measures of central tendency were calculated to describe the proportions of categorical variables, such as participants’ demographic characteristics and the response rate for each *Delphi* technique stage and preferred positioning. The definition of consensus for ranked data was: ≥75% accumulative votes within two more favourable (retained) or least favourable (rejected) categories on the 4-point polling and a mean of ≤1.75 or ≥3.5 on the 5-point Likert scale.

When opinions were broad, e.g., medicines capability, variances and *SDs* were affected by extremely high or low values on the 5-point Likert scale; central tendency measures did not capture where the majority sat on the topic [51]. Hence, the mean absolute deviation from the median (MADM) was calculated to measure participant disagreement and data spread, ignoring data outside of a trend [51]. The level of agreement was categorised into MADM thirds (low > 0.75, moderate 0.48–0.75, high < 0.48)—the average distance of participants’ ratings from the group’s median rating [51]. For items ranked according to medians; 4–5 was defined as strong, 3–3.5 as moderate, and 1–2.5 as weak support.

#### 2.8.3. Techniques to Enhance Trustworthiness

Triangulation helped produce more comprehensive findings by integrating information and perspectives from different sources and approaches, which were conferred with the EAG to overcome qualitative research’s intrinsic biases [52]. Approaches applied to check the meaning of the data generated from the study included reflexivity, peer-debrief, cross-coding, audit trail, and reflective journaling. Researchers regularly evaluated data collection methods on two composite processes: fidelity to the subject matter and utility in achieving research goals [53].

## 3. Results

Eighty-four expert diabetes health professionals participated in the study. The majority were registered nurses (*n* = 60 [71%]), ten of whom held midwifery qualifications, and the remainder were pharmacists (*n* = 7 [9%]) and allied health professionals (*n* = 17 [20%]) (see Appendix A). These experts resided in all Australian areas of remoteness and represented various healthcare sectors, including primary care, tertiary care, and academia.

### 3.1. STAGE I: Pre-Delphi Consultation

#### 3.1.1. Findings to Inform the Framework

*Self-regulating scope of practice:* Findings supported the need to explore further the scope of practice of diabetes educators and the different disciplines eligible for credentialling as a CDE [34]. Participants expressed concerns about the challenges associated with autonomous decision-making by individual health professionals for determining personal scope of practice. Participants indicated that the lack of clarity about scope of practice among diabetes educators from different disciplines created:Difficulties in identifying role boundary delineation between disciplines;Potential unsafe practices in diabetes care.

Many participants indicated that the exposure and training accessed by the health professional might not be adequate to provide safe care in certain circumstances when the expansion of scope is driven by health service needs rather than the individual’s skills or the wishes of consumers.

*Access to quality mentoring:* Findings suggested the need to incorporate a mechanism for ongoing mentoring in diverse diabetes areas [34]. Participants expressed concern about access to and quality of mentoring and indicated that access to the diversity of diabetes care is imperative. Most participants attributed differences in diabetes capabilities to the individual health professional’s access to lifelong mentorship in diabetes. Moreover, participants suggested deficits in competence derived from narrow mentorship. Participants described ‘narrow mentorship’ as relating to the mentors’ skills, knowledge, and exposure, whose experience may be limited to one diabetes population.

#### 3.1.2. Findings to Inform the Delphi Survey

The findings suggested that scope of practice, medicine management and non-medical prescribing, and diversity and training among health disciplines required exploration in the survey [34].

#### 3.1.3. Findings for Policy Advice

*Drugs and Poisons Legislation and terminology used:* Findings confirmed the need to improve consistency across *Drug and Poisons Legislation* about how health professionals can be involved in medicine management. Participants expressed concerns about the lack of consistency between Australian States and Territories legislation, which hampered mentoring. Often this related to improved clarity about the medicine management practices non-medical health professionals can support or amendments to medications once prescribed. Participants recommended consistency in legislation nationwide.

*Non-medical prescribing supporting healthcare access:* Findings suggest non-medical prescribing is a workforce solution. Participants expressed that non-medical prescribing by health professionals knowledgeable about diabetes management and medicine could support an appropriate increase in, and timely access to, healthcare. They recommended considering an advanced level of CDE with a limited prescribing endorsement to increase capacity.

*Remuneration for private sector CDEs:* Findings suggest that the private sector CDE workforce, including nurse practitioners, requires improved Government remuneration to enable viable business models. Participants reported discrepancies in remuneration for disciplines providing diabetes education and care. They indicated that these inconsistencies prevented working full-time in private practice because a business would not be viable. Moreover, funding impediments hampered the growth of the diabetes workforce, which did not allow for private diabetes educator workforce growth to meet primary care needs [34].

### 3.2. STAGE 2: Delphi Survey

#### 3.2.1. Delphi Survey Phase I Findings

The response rate was high: 88% and 91.7% across the two rounds (see Appendix A). Analyses of Phase one responses and EAG consensus identified four models to describe health professional practice levels. Model 3 (multidisciplinary) and Model 4 (work-setting focus) had smaller *SDs* and variances than the other models, suggesting more congruency in positioning. Model 3 had the lowest mean of 1.95 (*SD* 0.81, variance 0.65) and an accumulative ranking of 79% when the two preferred positions were calculated and accepted as the consensus preferred model. Model 4 had the highest mean of 3.33 (*SD* 0.88, variance 0.78) and an accumulative ranking of 86% in the two least preferred positions. Model 1 achieved a mean of 56% and Model 2 a mean of 51.1% and larger variances (see Appendix A).

The multidisciplinary model included seven health professional diabetes practice levels within the workforce requiring a difference in diabetes training (see Appendix A) and aligned to a stage of *diabetes clinical competence* from foundational to master (see Appendix A). Practice levels one to three were generalist roles, and four to seven were diabetes-specific roles.

#### 3.2.2. Delphi Survey Phase 2 Findings

The response rate remained high across the two *Delphi* survey rounds: 83% and 80.4% (see Appendix A). Analysis and EAG consensus of participants’ responses about the knowledge, skills, and attributes carried over to Phase two identified nine capabilities required to deliver diabetes education and care (see Table 1). Notably, the capabilities differed in focus; practice levels one to three emphasised awareness and promotion, and levels four to seven exemplified adept advanced diabetes skills (see Table 1).

The analyses also identified three groupings of attributes that underpinned the diabetes capabilities required by health professionals (see Appendix A): attributes to support excellent communication, collaboration, and advocacy; strive for excellence; and ensure the professional’s health and wellbeing to enable adaptability in dynamic environments.

Overall, 257 capability statements were accepted in *Round one*. All statements listed under practice levels four to seven reached a consensus. Five statements reached 74% and were also accepted (see Appendix A for ranking scores). Twenty-eight statements did not reach consensus derived from practice levels one to three and were neither accepted nor rejected. The strength of agreement was high, i.e., MADM <48 for most capability statements in practice levels four to seven. However, there was more disagreement related to capability statements for practice levels one to three in the following capabilities: clinical assessment capacities, DSME, counselling, technology, and QUM, where the strength of agreement was low, i.e., MADM >75 (see Appendix A).

In *Round two*, six capability statements gained consensus; four were accepted, one was rejected, and following MADM calculations, one was removed from the list based on the strength of disagreement. The last 22 capability statements carried over to Stage III.

### 3.3. STAGE III: Post-Delphi External Appraisal

Focus group participants accepted nine capability components and twelve more after altering the emphasis to ‘awareness’ or ‘promoting’ the issue to prompt action, and one was rejected. Participants encouraged the addition of Pharmacy Assistants and requested a capability component relating to disability, added in practice level one. They identified how colleges and universities could use the framework for training and how employers could use the framework to identify skill mix requirements. They emphasised that it was not a job description but a platform to guide training to better care for people with diabetes.

External health profession appraisal was largely favourable from the ten responders (31.3%). The majority indicated that the ‘Capability Framework’ would support developing a competent, flexible, and adaptive workforce for diabetes care and would not impede practice. Advice included adding the midwife more frequently and reconsidering the term generalist to prevent impeding practice. Most indicated the framework would not lead to unsafe practice; however, they advised reconsideration of an unrealistic medicine capability for healthcare assistants. Following EAG amendment advice and consensus, the *Capability Framework for Diabetes Care* was finalised (see the final allocation of roles in Appendix A).

## 4. Discussion

The findings have both workforce and policy implications that can increase capacity within the workforce for diabetes care if adapted.

### 4.1. Workforce Implications

#### 4.1.1. Workforce Diabetes Health Literacy and Preparedness for Diabetes Care

To be enabling, *health environments and the workforce must be health literate*—explicitly to support people living with a chronic condition due to the complexity and fragmentation of the healthcare system and health professionals’ impact on consumer health literacy [3,54]. The study highlighted the importance of a diabetes health literate workforce to build capacity and identified a guiding ‘Capability Framework’ [34]. Increasing workforce capabilities in diabetes care will increase the capacity for helping consumers to understand, appraise, and apply information to motivate and make effective decisions [55]. Currently, the health workforce cannot adequately support diabetes health literacy.

Outcomes of the *Diabetes Care Project* study, a Medicare-incentivised model, suggest that primary care nurses and allied health professionals’ skills and competence in managing diabetes currently may be inadequate to individualise diabetes care to support extensive clinical change [10]. The reduced primary endpoint HbA1c by 0.2%, although statistically significant in the study’s 5-point intervention group, was not clinically significant [56]. Similarly, a systematic review of international literature suggests that generalist/primary care health professionals lack confidence in knowledge and skills to individualise care and affect major changes in diabetes outcomes [14].

One reason for the lack of preparedness of the healthcare workforce for diabetes care may relate to health professional undergraduate courses. A national review of education about diabetes in undergraduate Bachelor of Nursing programs found Australian universities provided 4 to 21 h (0.5–2.5%) of diabetes-specific teaching across subjects [57]. Moreover, undergraduate students’ exposure to diabetes management during clinical practice varied, and the quantity was unmonitored [57]. Therefore, undergraduate health degrees may have inadequate diabetes content. Curricula require guidance about the level of knowledge and skills needed to improve health professional diabetes literacy and consistency, promoting consumer engagement [3,4].

A mismatch between priorities exists given the daily prevalence of people with diabetes accessing healthcare services in Australia [1] and diabetes’ national health priority status [13]. Without plans to assess or accredit undergraduate nurse courses regarding diabetes content [57], there is little prospect for a diabetes health-literate workforce [55]. It is unclear whether other professions’ undergraduate degrees have adequate diabetes content. National priorities are identified to create change to improve health outcomes; the ‘Capability Framework’ can guide relevant course content to support the priority. The ‘Capability Framework’ promotes a knowledgeable, compassionate health workforce, where, as a caring workforce, discussions about or with people living with diabetes is neither demeaning nor stigmatising, fostering engagement [3,4].

#### 4.1.2. Capability Framework for Diabetes Education and Care

The ‘Capability Framework’ guides the training of health professionals nationally. It uniquely supports whole workforce growth in diabetes consistently and creates shared understandings between disciplines. Moreover, the framework’s strength is its emphasis on capability-based learning, which focuses on outcomes [44,58,59], compared with existing competency-based diabetes frameworks. Rather than merely fulfilling the minimum requirements of a role, health professionals are encouraged to go beyond to reflect on learnt skills.

A recent American study using a 5-round modified-Delphi technique identified 130 competencies across six domains that align with the capability areas of the current study: clinical management practice and integration, communication and advocacy, person-centred care and counselling, research and quality improvement, systems-based practice, and professional practice [60]. Similarly, the competencies were intended to guide practice regardless of discipline; however, competencies were presented as health areas, e.g., healthy eating, for the Diabetes Care and Education Specialists only and not the broader health workforce, reducing adaptability and consistency across the workforce. The study also used a substantial sample—457 participants—which creates data analysis challenges to ensure accurate interpretation [40]. Furthermore, the final product was not validated with ‘real users’; the last Delphi appraisal rounds were not independent. 

Competency frameworks’ ties to one-dimensional views of learning, healthcare roles, and discipline requirements create limitations in a dynamic health system [58]. Moreover, national and international diabetes competency frameworks often focus on individual health topic areas or one health professional discipline [34,61,62,63,64,65], reducing flexibility and consistency in training and language across disciplines.

The findings from the current study are important because the application of Sen’s capability approach to the education setting has been found to increase flexibility and responsiveness, translating to increased workforce capacity [59]. Sen’s capability approach is a personal utilitarian approach with two specific elements: functioning and agency [66]. From an agency perspective, individuals feel happiest and perform the best when they are more satisfied by pursuing goals they value [59,66]. Combining Sen’s theory with career educational development approaches can create opportunity and increase capacity drawn from agency [44]. Promoting the concept of agency in education can educate health professionals to reason on personal decisions and preferences, enhance their capacities to reflect critically on their surroundings and work environment, envisage changes, and cultivate capacities to realise such changes in practice [66].

Evidence suggests that competence derives from a functional sphere—the functional facility to solve complex problems that turn a series of elements such as knowledge and skills into competencies, whereas capabilities derive from an ethical and normative sphere—the opportunity to choose an action, choice, or behaviour [66]. Thus, the capability approach is broader and not demand-orientated; health professionals are guided by the freedom to choose and develop, thus giving rise to autonomy [44,66]. Consistent with Sen’s capability approach, the framework promotes self-efficacy, focusing on health professionals’ capabilities and enabling targeted education as evidence evolves, leading to increased workforce capacity [59].

Real-time learning fosters the health professional’s ability to adapt to clinical situations [58]. Given this, making the ‘Capability Framework’ an online and ‘living’ framework has merit because it can be updated as evidence evolves, which aligns with the *Australian National Diabetes Strategy* that recognised the need for increased preparedness [13]. A ‘living’ framework creates an opportunity to enable increased workforce adaptability, including accelerating change when faced with emerging public health challenges.

The ‘Capability Framework’ promotes multidisciplinary diabetes credentialling and recognises diversity through including diverse disciplines in the diabetes practice levels, increasing the capacity for diabetes care in the workforce. The ‘Capability Framework’ assists workforce planning when used as a tool to identify staff working at each practice level in an organisation or region, thereby identifying workforce gaps. Furthermore, it can support capacity-building by integrating different disciplines and expertise within diabetes teams, guiding more comprehensive diabetes assessment and care.

A limitation of multiple frameworks, as used in the United Kingdom and Australia, to guide different disciplines in diabetes is inconsistency in training across professions [9,61,62,63,64]. Moreover, there is a risk of reduced relevancy because diverse organisations’ priorities drive ongoing reviews. Discipline-specific competency-based training might not enable expertise because of the ties to specific workplace healthcare roles and discipline requirements rather than accommodating for the dynamic changes in healthcare and the person with diabetes needs [58]. Thus, diversity in diabetes teams combined with capability-based training can promote more in-depth clinical reasoning using cross-discipline learning. The ‘Capability Framework’ sets standards for diabetes education and care, which could, in turn, reduce role boundary confusion by clearly identifying the requirements and capabilities for diabetes education courses.

#### 4.1.3. Mentoring and Practice

The findings suggest that lifelong quality mentoring is critical to improving practice and that the geographic maldistribution of different health professionals impacts their access to quality diabetes mentoring [34]. An approach is required to support quality mentoring and stronger links between newly graduated and generalists with diabetes-competent health professionals to build the capacity for diabetes care within the healthcare workforce [15,16,17,18]. Moreover, access to two-way mentoring relationships between Aboriginal health workers and non-Indigenous health professionals is needed to promote healthcare access equality in Australia [67]. Internationally as in Australia, significant disparities in Indigenous populations’ socioeconomic status and environmental contexts inextricably result from past and contemporary colonialism policies and practices that continue to drive inequities that have persisted for generations [23,24,67]. Mentorship and working with Indigenous health workers can reduce fear in the Indigenous person accessing care and promote safe cultural practices [23,24,67,68].

The ‘Capability Framework’ can guide both mentoring and clinical practice. Ericsson identified that the acquisition of expert performance comes from ongoing training, mentoring by others, and reflection in and on practice to improve approaches to undertaking an activity [43]. To move towards expertise, Benner’s stages of clinical competence suggest that the health professional moves through stages where concepts are tested [42]. Combining these theories created a framework to ensure more health professionals develop expertise in diabetes to meet consumer needs.

As with other countries, barriers exist in Australia, nuanced by remoteness, poor dispersion, and access to suitable mentors. Therefore, the ‘Capability Framework’ requires the addition of a mechanism or technology to allow remote and rural CDEs to access quality mentors routinely to build capacity in diabetes care.

### 4.2. Policy Implications

#### 4.2.1. Medicine Management in Diabetes Care

The ‘Capability Framework’ provides policy and guidance for organisations about how health professionals at different practice levels can support medicine management. Further, it articulates a structure to describe a career pathway for diabetes educators to increase their scope of practice into non-medical prescribing. However, only nurse practitioners can prescribe, and appropriately endorsed podiatrists and pharmacists, who may be a CDE, have limited prescribing rights in Australia. Therefore, consideration should be given to developing an appropriately trained advanced-level CDE role focused on complex care, with prescribing endorsement against a glucose-lowering formulary as a workforce solution to increase capacity for diabetes care.

Developing diabetes medicine management capabilities within the health workforce is crucial, primarily due to high rates of poor diabetes medicine adherence and increasing morbidity and mortality [68,69]. A recent study following 1.2 million Australians living with type 2 diabetes found inequities; those residing in remote or disadvantaged socioeconomic groups were less likely to access newer medicines with cardiovascular benefits than those in metropolitan areas [70].

Promoting medicine management capabilities and non-medical prescribing has increased access to appropriate care, which has produced economic benefits and improved patient outcomes [39]. Findings suggest CDEs already work collaboratively with medical officers who often seek their advice on diabetes medicine—an underutilised expert resource with current knowledge about glucose-lowering medicines [34]. Identifying a means for supporting non-medical prescribing capabilities using experienced and skilled CDEs could promote increased care in remote areas.

However, the study revealed that CDEs do not share an understanding of how they should be involved in medicine management because of the profession’s multidisciplinary nature and multiple legislation [34]. The lack of shared understanding creates confusion, differences in practices and what the consumer can expect to receive, or unsafe practices by inadequately trained health professionals. Current national registration provides a mechanism for one national *Australian Drug and Poisons Legislation* that could reduce inconsistencies in training across the nation. Legislation must include information to explain whether and who can amend a prescriber’s order once the medication is prescribed and dispensed—also, a term to describe these medication amendments consistently.

#### 4.2.2. Promoting Viability of Private Practice Workforce for Diabetes

Improving private CDEs remuneration and identifying opportunities to enable nurse practitioners to develop, implement, and evaluate *Chronic Condition Management Plans* in partnership with people living with diabetes can create workforce solutions to increase timely quality healthcare access and build workforce capacity. International research has established that nurse practitioners match or exceed physician colleagues in providing quality care in primary care and speciality areas [71]. However, according to Australian nurse practitioners, the current structure of the Pharmaceutical Benefits Scheme and Medicare restricts their ability to deliver a complete cycle of care [72], reducing workforce capacity.

An imperative exists to sustain timely care, as approximately 12% of Australian endocrinologists and diabetes physicians live outside major cities [25,26]. In contrast with Australia, the UK and USA diabetes educator workforce have infiltrated primary care [8,73,74]. In the UK, although 49% work in tertiary settings, 36% of diabetes nurse specialists are in the community, and 15% are across both; also, they often hold non-medical prescribing endorsement [74]. Currently, 44% of the population has some level of privately funded health cover [75]. Hence, Australia’s capacity for preventative care can be increased by supporting the growth of private sector CDEs and nurse practitioners. The growth of these roles in the private sector builds workforce capacity. They can support promoting medication adherence in areas of increasing remoteness where access or adherence is low [26,68,69] and can increase screening and early intervention [72].

#### 4.2.3. Orientating Healthcare Organisations for Capability-Based Learning

Capability-based learning will help to increase workforce capacity in healthcare organisations and *enable compassionate, high-quality healthcare*. Healthcare organisations need to promote health professional autonomy to enable capability-based learning, which fosters innovation [59]. Growth and opportunity may be impeded if employers and organisations are not conducive to allowing autonomous practice on which capability-based learning is based [36]. Social conditions and arrangements in organisations need to promote autonomy and initiative because knowledge and skills can become obsolete in a rapidly evolving world, so it is essential to focus on supporting traits that make people perform [76]. Given that individual performance is influenced by *self-concept* and *motivation*, supporting autonomous practice provides more satisfaction and is more likely to promote these traits [76].

### 4.3. Limitations

Despite the proportion of disciplines sampled aligned with those from the ADEA membership survey, which suggests the sampling frame would comprise primarily nurses with one-fifth from pharmacy and allied health disciplines [8], some disciplines were not represented in the study. Only a few physiotherapists and Indigenous health practitioners were CDEs and did not meet the inclusion criteria because these professions only became eligible for CDE credentialling after 2015 [77,78]. Both groups play an integral role in the healthcare system and Indigenous health in many parts of Australia; lack of feedback was a limitation. However, further work has promoted their involvement in designing and implementing an online *Capability Framework for Diabetes Care*.

Other limitations were the level of information required in the first round of *Delphi* survey Phase one and the sheer volume of competency statements for *Delphi* survey Phase two ranking. The *Delphi* technique requires adequate time and participant commitment through the whole process to ensure the data collected are truly representative of the expert panels’ opinions [79]. An inherent risk was that *Delphi* participants would not read each capability statement and make considered ranking decisions or that they would drop out. Researchers reviewed the length of time taken to undertake the survey and scores for similar ranking patterns in case panellists did not engage positively with the survey. The length of time it took to undertake the survey and differences in ranking within ‘contentious’ capabilities were quite diverse, implying participants engaged positively in the process.

## 5. Conclusions

Policy direction must be supported by the alignment of appropriate, consistent diabetes healthcare workforce training. A rigorous process of consultation and consensus developed the *Capability Framework for Diabetes Care*, which addresses workforce enablers identified by the *Australian National Diabetes Strategy.* The framework establishes diabetes-related practices and capabilities to guide and improve diabetes health care advice and delivery in the workforce. A better diabetes-prepared health workforce might increase early detection of diabetes and complications, improve referral processes, improve care access, and reduce the risk of complications and associated healthcare costs.

Consumer engagement is supported when health messaging is consistent, and the ‘Capability Framework’ helps establish consistent health professional curriculum development across the nation. By promoting multidisciplinary credentialling and recognising diversity, the framework consistently supports the workforce’s development to create shared understandings of diabetes and evidence-based care across health professional disciplines. The framework addresses social and economic diabetes healthcare gaps by guiding remote area health professionals. Consistent terms to describe diabetes-related medicine management and prescribing and clarity about how health professionals can manage medicines once prescribed are necessary to improve healthcare for people with diabetes. 

Future studies should evaluate the impact of the ‘Capability Framework’ if implemented nationally and whether specific capabilities are required to better tailor care to Indigenous people and the feasibility and viability of nurse practitioner-driven *Chronic Condition Management Plans* for people with diabetes. The findings from this study will inform diabetes policy, practice, education, and research.

## Figures and Tables

**Figure 1 ijerph-19-01012-f001:**
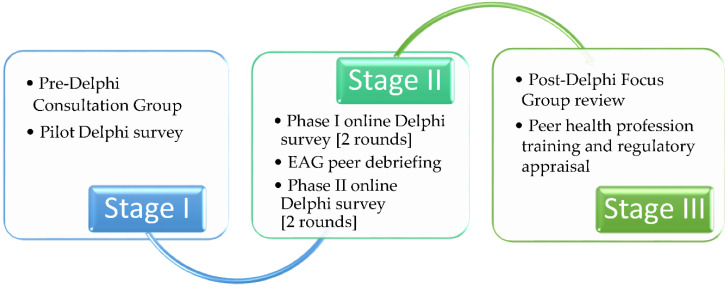
Stages for modified Delphi technique.

**Figure 2 ijerph-19-01012-f002:**
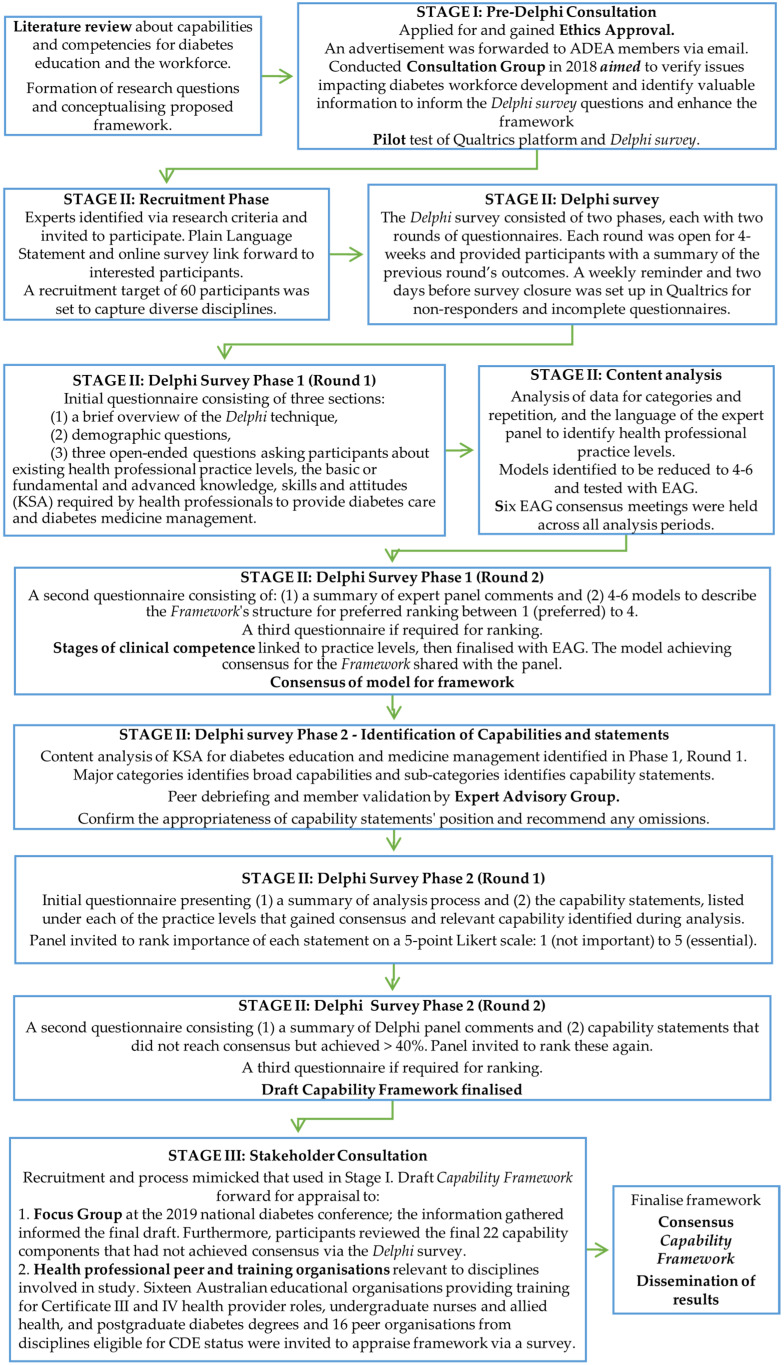
Flow chart guiding chronology of data collection methods.

**Table 1 ijerph-19-01012-t001:** Nine capabilities required to deliver and support diabetes care and their definition.

**Diabetes capabilities for practice levels one to three.** The capabilities focus on awareness or promotion and are defined as:
*Displays clinical assessment capacities:* demonstrates foundational skills and is developing clinical assessment skills relevant to diabetes to assist with diagnosis and care and identify health status changes.
*Supports diabetes self-management education:* promotes, assists, and encourages self-management considering the needs, goals, and life experiences of the person with diabetes using teaching skills and a structured decision-making process guided by evidence and an understanding of health literacy.
*Builds therapeutic relationships:* develops intentional connections focused on person-centeredness and shared decision-making between a healthcare professional and an individual requiring diabetes support; the positive relationship engaged for effecting a beneficial change towards an individual’s goal.
*Communicates with influence:* informs with the intention to affect behaviour change; to inspire, motivate, encourage, guide, and advocate for people living with diabetes or prediabetes.
*Supports counselling to achieve the best outcomes:* uses supportive counselling techniques within an empowerment framework when guidance on actions is required and identifies mental health issues in people living with diabetes, such as diabetes-related distress or burnout and depression.
*Supports quality use of medicines (QUM):* demonstrates QUMs in a supportive role and identifies potential medicine-related risks and benefits.
*Displays quality use of diabetes technology:* demonstrates a supportive role with individuals using technology and identifies potential risks.
*Supports care coordination:* assists in care coordination and transition as directed or is involved in developing and implementing a care plan for the management of diabetes and facilitates appropriate services in conjunction with a medical team.
*Achieves quality:* displays a supportive role within quality and research activities and incorporates evidence in all practice elements.
**Diabetes capabilities for practice levels four to seven.** The capabilities focus on diabetes healthcare professionals who work at an advanced practice level and possess adept diabetes skills in diabetes care and education, and the capabilities are defined as:
*Exemplifies clinical assessment capacities:* advanced assessment skills and knowledge, clinical acumen, and reflection in and on practice to enable more comprehensive and individualised assessment for the person with complex diabetes issues and accumulative comorbidities.
*Shapes diabetes self-management education (DSME), support and care:* competent to design, implement, deliver, and evaluate structured DSME. Influences the continuous development of improved diabetes care skills in consumers and practices to optimise DSME, support, and care. Innovative and leads changes; promotes and develops processes to support improvement in health literacy.
*Builds therapeutic relationships:* adept at developing positive relationships between healthcare professionals and individuals requiring diabetes support through structured shared decision-making. Mentors others and incorporates evidence into local protocols, guidelines, and the organisation to better engage with people living with diabetes and their carers and effect beneficial change.
*Communicates with influence and leadership:* proficient at communicating with the intention to achieve an effect: to listen, inspire, motivate, and encourage both the consumer and other healthcare professionals. Leads with purpose and promotes wide-reaching advocacy for people living with diabetes.
*Exemplifies counselling to achieve the best outcomes:* adept at using supportive and empowering counselling techniques and implements significant evidence into guidelines, protocols, and the organisation. Proficient at detecting mental health issues early, such as diabetes-related distress, diabetes burnout, and depression.
*Exemplifies quality use of diabetes technology:* proficient or regarded as an expert at teaching, operating, and monitoring diabetes technology via different processes, i.e., providing self-management education, technology advice and care, and, in some cases, prescribing if the device administers a medicine.
*Exemplifies QUMs:* proficient or regarded as an expert at ensuring diabetes medicines are used safely when needed. Delivers comprehensive medicines management, including essential elements such as rigorous monitoring and de-prescribing when indicated.
*Leads care coordination:* adeptly coordinates relevant stakeholders involved in the consumer’s care to deliver appropriate healthcare services promptly and efficiently. Aware of the specific requirements of vulnerable groups and people with complex diabetes issues and monitors and evaluates outcomes.
*Cultivates quality through leadership and research:* identifies, engages in, mentors, and leads research and quality and safety improvement activities to identify ways to improve diabetes education and care.

## Data Availability

The data presented in this study are available on request from the corresponding author. The data are not publicly available due to ethical reasons.

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
