# Peer review of "Diabetes Capabilities for the Healthcare Workforce Identified via a 3-Staged Modified Delphi Technique"

_ijerph, 2022, doi:10.3390/ijerph19021012_

Round 1
Reviewer 1 Report
Congratulation for the initiative. The research cover an interesting area that needs more research for the repercussions in daily life of people living with diabetes.
The text describes the methodology used well but it is extensive and can cause distraction for the reader. I suggest reducing the length of the Methods section to focus more on Discussion. A flowchart can be used to describe the process and reduce the text.
The discussion section also has a lot of text that can be abbreviated. In some paragraphs the message to be conveyed overlaps with other paragraphs and distracts from the central theme. I recommend trying to be shorter and synthesize the concepts, be more specific.
Author Response
Thank you for your comments. The methodology section has been revised, including directing the reader to Figure 2 that explains the methods used. The authors have retained some information about the methodology because research articles identified during the literature search for this study often provided very little detail, making it more challenging to understand the steps taken and to replicate the method. Further, paragraphs with overlapping messaging in the health literacy and mentoring sections have been synthesised.
Reviewer 2 Report
The authors present their studies on the development of a consensus capability framework to guide non-medical health professional training in delivering diabetes education across the Australian healthcare system. The topic are very significant and the paper are well written, I just have a few minor comments for the authors:
- The introduction is a bit lengthy with a lot background information of diabetes in general, I would suggest the authors to focus on the current problem and the objectives of this study.
- There were two paragraph labeled as 2.3, please revise.
Author Response
Thank you for your comments. Information about diabetes generally has been removed, and only information concerning the workforce and the policies that funds workforce, and capacity building has been retained. The numbering of paragraphs has been corrected.
Reviewer 3 Report
Thank you very much for your hard work in performing this very interesting study. I think this study will be helpful for diabetics.
I wrote some comments below. I hope it will help you revise your manuscript.
Introduction section
- please describe the previous study on the modified Delphi technique.
Conclusions
-Please suggest what future study should be done.
Author Response
Thank you for your comments. No modified or straight Delphi technique studies were identified in the literature search to identify diabetes workforce capabilities. However, an example of a previous modified Delphi study for workforce development has been discussed in the discussion section, reference 60 on page 12. Further, areas of future study have been added to the conclusion.